# Disease burden of ageing, sex and regional disparities and health resources allocation: a longitudinal analysis of 31 provinces in Mainland China

Shu Chen [1,2] Yafei Si [1,2] Katja Hanewald [1,2] Bingqin Li [3]
Hazel Bateman [1,2] Xiaochen Dai [4,5] Chenkai Wu [6] Shenglan Tang [6,7]

For numbered affiliations see end of article.

**Correspondence to**
Dr Shenglan Tang;
shenglan.tang@duke.edu

## ABSTRACT

**Objectives** To measure the disease burden of ageing based on age-related diseases (ARDs), the sex and regional disparities and the impact of health resources allocation on the burden in China.

**Design** A national comparative study based on Global Burden of Diseases Study estimates and China's routine official statistics.

**Setting and participants** Thirty-one provinces of Mainland China were included for analysis in the study. No individuals were involved.

**Methods** We first identified the ARDs and calculated the disability-adjusted life years (DALYs) of ARDs in 2016. We assessed the ARD burden disparities by province and sex and calculated the provincial ARD burden-adjusted age. We assessed historical changes between 1990 and 2016. Fixed effects regression models were adopted to evaluate the impact of health expenditures and health workforce indicators on the ARD burden in 2010–2016.

**Results** In 2016, China's total burden of ARDs was 15 703.7 DALYs (95% uncertainty intervals: 12 628.5, 18 406.2) per 100 000 population. Non-communicable diseases accounted for 91.9% of the burden. There were significant regional disparities. The leading five youngest provinces were Beijing, Guangdong, Shanghai, Zhejiang and Fujian, located on the east coast of China with an ARD burden-adjusted age below 40 years. After standardising the age structure, western provinces, including Tibet, Qinghai, Guizhou and Xinjiang, had the highest burden of ARDs. Males were disproportionately affected by ARDs. China's overall age-standardised ARD burden has decreased since 1990, and females and eastern provinces experienced the largest decline. Regression results showed that the urban–rural gap in health workforce density was positively associated with the ARD burdens.

**Conclusion** Chronological age alone does not provide a strong enough basis for appropriate ageing resource planning or policymaking. In China, concerted efforts should be made to reduce the ARDs burden and its disparities. Health resources should be deliberately allocated to western provinces facing the greatest health challenges due to future ageing.

---

## STRENGTHS AND LIMITATIONS OF THIS STUDY

⇒ This study uses longitudinal data from 1990 to 2016 in 31 provinces of Mainland China to assess the age-related disease (ARD) burdens.

⇒ The study generates high-quality evidence on disparities in ARD burdens across provinces, sexes and disease categories in China.

⇒ The study adopts robust fixed effects regression models to assess the impact of health expenditures and human resources for health indicators on the ARD burden.

⇒ Although the study has used the best data sources available for the analysis, future research could generate more up-to-date results once more recent data become available.

---

## INTRODUCTION

The world is rapidly ageing due to increased life expectancy and decreased fertility rates.[1] Health is a critical factor in determining whether population ageing means more opportunities or challenges to society.[2] Although lifespans have increased substantially worldwide, it is unclear whether healthspans, that is, healthy and morbidity-free lifespans, have improved similarly.[2–9] It is, therefore, essential to measure the interactions between ageing and health for a clearer understanding of the disease burden of ageing. Numerous metrics have been developed to measure population ageing and health for resource planning. Traditionally, the change of chronological age and age structure have been used to assess ageing based on demographic metrics such as life expectancy or the percentage of the population over a certain age threshold.[10] Another set of metrics assesses functional status through indicators such as frailty,[11 12] disability,[13–15] cognitive function[16] and intrinsic capacity.[17 18] Health status is typically estimated based on biomarkers, self-reported health status[19 20]

and epidemiological indicators of different diseases such as incidence, prevalence, mortality and disability-adjusted life years (DALYs). Some metrics measure ageing while also taking health status into account. These metrics include healthy life years, healthy life expectancy (HALE) and biological age. However, these metrics either measure ageing and health separately or cannot provide disease-specific burden information to guide resource planning.

To complement the existing metrics and measure the population-level disease burden of ageing, Chang *et al* developed a novel metric called the age-related disease (ARD) burden.[21] They defined ARDs as diseases with incidence rates that increased quadratically with age. The researchers used estimates from the Global Burden of Diseases, Injuries, and Risk Factors Study (GBD) 2017 to measure and compare ARD burdens of 195 countries from 1990 to 2017. They found that non-communicable diseases (NCDs) were the primary contributor to ARDs, and there were significant variations in ARD burdens across countries. Of the 195 countries, Switzerland had the lowest ARD burden, while China ranked 75th and Vanuatu had the highest. The researchers found that the age-standardised ARD burden dropped globally between 1990 and 2017 due to lower disease severity and case fatality rates. Hu *et al* assessed China's ARD burden based on the GBD 2017 Study estimates at the national level. The researchers consistently found that NCDs are the primary contributor to ARDs in China. The age-standardised ARD burdens also decreased between 1990 and 2017, and the magnitude of ARD decrease was larger among women than men.[22] These findings emphasise the inadequacy of using chronological age alone to inform resource planning and policy design. They confirm the importance of considering health status and disease severity within the context of ageing and the need for concerted efforts to address regional disparities, especially for regions or countries with significant development inequalities.[21 22]

Many previous studies have assessed whether and how health resource allocation can impact health outcomes.[23–29] Health workforce density and total health expenditures per capita have been widely used as key proxies for health resources. A large body of high-quality evidence supports the view that a higher health workforce density can help improve health outcomes such as life expectancy, infant mortality rate, under-5 mortality rate and maternal mortality rate.[23 24] Evidence further shows heterogeneity among different types of the health workforce in terms of how they impact health outcomes.[23 24] The impact of health expenditures on health outcomes is mixed.[26–29] For example, evidence shows that an increase in health expenditures per capita can positively impact health outcomes, including life expectancy, under-5 mortality and maternal mortality in sub-Saharan African countries.[29] However, a similar study did not find supporting evidence in Organisation for Economic Co-operation and Development countries.[26] Another study found a significant association between higher health expenditures per capita and lower infant mortality in 15

European Union countries, but only marginal increases in life expectancy.[28]

China is a large, rapidly ageing country with uneven economic and health development across provinces. In 2020, 264 million of China's inhabitants (18.7%) were over 60 years, ranking first globally by this metric, and this number is projected to double by 2050.[1 30] Generally, in China, the economy is most developed in eastern provinces, followed by the central and western provinces. For example, in 2020, the gross domestic product (GDP) per capita in Shanghai, a municipality located on China's east coast, was 4.3 times higher than that of Gansu, a province in western China.[30] Regional disparities in health development are similar in scale; provinces in the eastern region generally perform best, followed by the central and western provinces.[31] For example, among males in 2015, the average HALE was estimated to be around 78 years in Beijing, Tianjin and Shanghai, compared with 69 years in Qinghai, Tibet and Yunnan, located in western China.[32] China is estimated to attain 12 of the 28 health-related Sustainable Development Goals indicators by 2030, and the country's eastern provinces are estimated to achieve a higher number.[33] Therefore, understanding the disparities in the disease burden of ageing across provinces and the impact of health resource allocation on the burden is crucial for China's central and provincial governments to plan for an ageing society. Nevertheless, no study has yet been conducted for this purpose. To fill the evidence gap, this study aims to: (1) measure the burden of ARDs in China at the subnational level, examining disparities across provinces and by sex, (2) assess changes in ARD burden from 1990 to 2016 and (3) explore how health resource allocation impacts the ARD burden. We also examined the difference by disease group whenever possible.

## METHODS
### Overview
This study used estimates from the GBD Study to calculate and analyse the ARD burden in China. The GBD team have updated their estimates annually since 2015 (except for 2018), with improvements on disease causes, risk factors, data input sources and modelling strategies to obtain updated and robust estimates. The most recent publicly released estimates were GBD 2019.[34 35] We used GBD 2019 Study estimates to select ARDs and calculate the total national burden considering its timeliness. We analysed the ARD burden at the subnational level based on estimates of the GBD 2016 Study, as they were the most recent subnational estimates of China our team could access. However, we did not find significant changes in the causes and estimates that could impact the robustness of the study findings after careful comparison. The detailed methodology of the GBD 2016 and 2019 Studies has been published elsewhere.[35 36] The GBD Study classified the causes hierarchies into four levels, each with increased specificity.[37] Level One consists of three general

causes: communicable diseases, NCDs and injuries. Levels Two, Three and Four further divide these causes into subgroups.[37] The data related to provincial health expenditures, human resources for health and economic and demographic indicators were obtained from China Statistical and Health Statistical Yearbooks, providing data from 2010 to 2016 for regression analysis.

The geographical units of analysis are provincial-level jurisdictions in Mainland China, including 22 provinces, 5 autonomous regions and 4 municipalities (hereafter referred to as 31 provinces).[30] Our study complies with the Guidelines for Accurate and Transparent Health Estimates Reporting (known as GATHER) statement, and we provided the GATHER checklist in the online supplemental appendix.

### ARD selection and burden measurement

Considering data availability, we followed the definition developed by Chang *et al* to measure ARDs as diseases with incidence rates that increased quadratically with age among the adult population.[21] Our selection of diseases focused on GBD Level Three causes, and the detailed selection methods have been published elsewhere.[21] To include a comprehensive list of diseases, we focused on the population aged 15 years and older.

The ARD burden was measured by calculating the sum of the DALYs of the identified ARDs in the entire population. DALYs are defined as the sum of the years of life lost due to premature mortality and the years lived with a disability due to a disease or health condition. We calculated the provincial ARD burden of NCDs to approximate the total ARD burden to analyse regional disparities, as previous studies showed NCDs contributed to over 80% of the ARD burden,[21 22] and we could only access the subnational estimates of NCDs. We reported the ARD burden rate, measured as DALYs per 100 000 population, to facilitate comparisons. To explore the burden composition by disease category, the ARD burden was further stratified and analysed at the second level of GBD causes within NCDs and by sex when appropriate. We also calculated the percentage decrease in the ARD burden to assess the historical change and clustered the results by disease category and region for comparative analysis (please see online supplemental table S1 for the administrative divisions).

The ARD burden-adjusted age was defined as the equivalent age measured by the ARD burden. We estimated the ARD burden-adjusted age of each province to facilitate an intuitive assessment and comparison across provinces. We first identified the national ARD burden-adjusted age based on its age-specific burden rate. We selected the two 5-year age groups with the closest burden rate as the national average and calculated the national average age by assuming a linear increase in the ARD burden within each 5-year age group. Afterwards, the provincial ARD burden-adjusted age was calculated, assuming each province shared the same burden rate per age as the national average (online supplemental appendix for calculation

details). A younger ARD burden-adjusted age, therefore, implied a lower ARD burden.

To compare the historical change in the ARD burden, we applied age standardisation to the calculation using the Institute for Health Metrics and Evaluation standard population age structure when necessary (online supplemental table S2 for the standard population structure). The 95% uncertainty intervals (UI), generated by the GBD Study team as the 25th and 975th ordered 1000 random draw values of the posterior distribution,[36 38] were also presented for the ARD burden estimates.

### Analysing the impact of health resource allocation on the ARD burden

We explored the impact of health resource allocation on the ARD burden using a panel data analysis approach. All equations were estimated with a log-linear functional form to enable unit-free comparisons of coefficients. To understand the underlying reasons for burden shifts over time, we used the age-standardised ARD burden rate as the dependent variable. Health expenditures and workforce density were adopted as proxy measures for health resources. Total health expenditure per capita was the key independent variable used to measure health expenditures.[26 27] The key independent variables used to measure health workforce density were three separate sets of indicators: total health professional density, licensed doctor density and licensed nurse density, all per 1000 population. Health professionals included licensed doctors (clinical, dental, public health and traditional Chinese medicine), licensed nurses, pharmacists, clinical laboratory technicians and radiologists.[39] We chose three sets of indicators to measure health workforce density because (1) together they accurately represent the distribution of China's health personnel resources; (2) they are widely used in published literature, and heterogeneity exists in terms of their impact on health outcomes;[23 24] and (3) data stratified by province and by urban and rural areas are available for all three indicators. We ran three separate regression models to include the three health workforce density indicators.

We included GDP per capita, sex and education as covariates in the regression models to account for the major socioeconomic determinants of the population health burden. China has made remarkable achievements over the past decade in reducing the health disparities between urban and rural residents, primarily through improving maternal and child health and extending health insurance coverage, among others, for its rural residents.[40] However, there is still a noticeable urban–rural gap in health development, including access to quality health services, health workforce quantity and quality and health outcomes.[41–43] Therefore, to account for the significant gaps in urban–rural development and health across China, we included the percentage of people residing in urban areas (to measure the process of urbanisation) and the ratio of urban–rural health workforce density in the model as covariates. The model also

included time dummies (to control period effects), province fixed effects and an error term (see online supplemental appendix for model details). We further explored the correlation of key variables to assess whether multicollinearity could undermine the robustness of the estimates for specific variables in the regression model (online supplemental tables S3 and S4). Standard errors were clustered at the provincial level. Province was the unit of analysis, and a fixed effects estimator was used to remove the potential endogeneity from time-invariant omitted variables.

Since provincial-level data on health expenditures per capita and health workforce density were only broadly available from 2010 onwards, our panel dataset included data from 2010 to 2016. We performed log-linear interpolation to obtain annual estimates of the ARD rate from 2010 to 2016, as the GBD Study only provides estimates in 5-year intervals.[38] The health expenditures per capita in 2010 are missing for seven provinces: Shanghai, Hainan, Sichuan, Tibet, Shaanxi, Qinghai and Ningxia. Only data on Tibet is missing in 2011. Therefore, after testing the robustness of the linear increase assumption, we imputed the missing data, assuming a linear increase in per capita health expenditures between 2010 and 2016. All analyses were performed in Stata V.16.0 (Stata Corp LLP, College Station, Texas, USA).

## Patient and public involvement statement

Patients or the public were not involved in the design, conduct, reporting or dissemination of this research.

## RESULTS

A total of 58 causes out of 169 causes (34.3%) at level three were selected as ARDs in China (table 1). In total, 9 of the 58 causes were infectious diseases (15.5%), 47 were NCDs (81.0%) and 2 were injuries (3.5%). In 2016, the total burden of ARDs in China was 15 703.7 DALYs (95% UI: 12 628.5, 18 406.2) per 100 000 population. NCDs accounted for 91.9% of the total burden. Among the NCDs, cardiovascular diseases (CVDs), neoplasms, chronic respiratory diseases (CRDs), sense organ diseases and neurological disorders were the five leading contributors to the ARD burden.

Significant regional and sex disparities in ARD burdens existed in China. Generally, the eastern provinces had the lowest crude burden rates, while the northeastern and several western provinces bore the highest ones (figure 1).

**Table 1** Summary of age-related diseases in China by category

| GBD causes at Level 1 | GBD causes at Level 2 | GBD causes at Level 3 |
|---|---|---|
| Infectious diseases | Respiratory infections and tuberculosis | Tuberculosis, diarrheal diseases, lower respiratory diseases |
| | Other infectious diseases | Meningitis, encephalitis, tetanus, varicella and herpes zoster |
| | Neglected tropical diseases and malaria | Cysticercosis, trachoma |
| NCDs | Neoplasms | Oesophageal cancer, stomach cancer, liver cancer, tracheal, bronchus, and lung cancer, prostate cancer, colon and rectum cancer, lip and oral cavity cancer, other pharynx cancer, gallbladder, and biliary tract cancer, pancreatic cancer, malignant skin melanoma, non-melanoma skin cancer, bladder cancer, brain and central nervous system cancer, mesothelioma, Hodgkin's lymphoma, non-Hodgkin's lymphoma, multiple myeloma, leukaemia, other malignant neoplasms |
| | Cardiovascular diseases | Ischaemic heart disease, stroke, hypertensive heart disease, cardiomyopathy, myocarditis, atrial fibrillation and flutter, peripheral artery disease, endocarditis |
| | Chronic respiratory diseases | Chronic obstructive pulmonary disease, asthma and interstitial lung disease |
| | Digestive diseases | Paralytic ileus and intestinal obstruct, vascular intestinal disorders, pancreatitis |
| | Neurological disorders | Alzheimer's disease and other dementias, Parkinson's disease, idiopathic epilepsy, other neurological disorders |
| | Diabetes and kidney diseases | Acute glomerulonephritis, chronic kidney disease |
| | Skin and subcutaneous diseases | Fungal skin diseases, pruritus, decubitus ulcer and other skin and subcutaneous diseases |
| | Sense organ diseases | Age-related and other hearing loss, other sense organ diseases, blindness, and vision loss |
| | Musculoskeletal disorders | Low back pain |
| Injuries | Unintentional injuries | Falls, drowning |

Data source: GBD 2019.
GBD, Global Burden of Diseases; NCDs, non-communicable diseases.

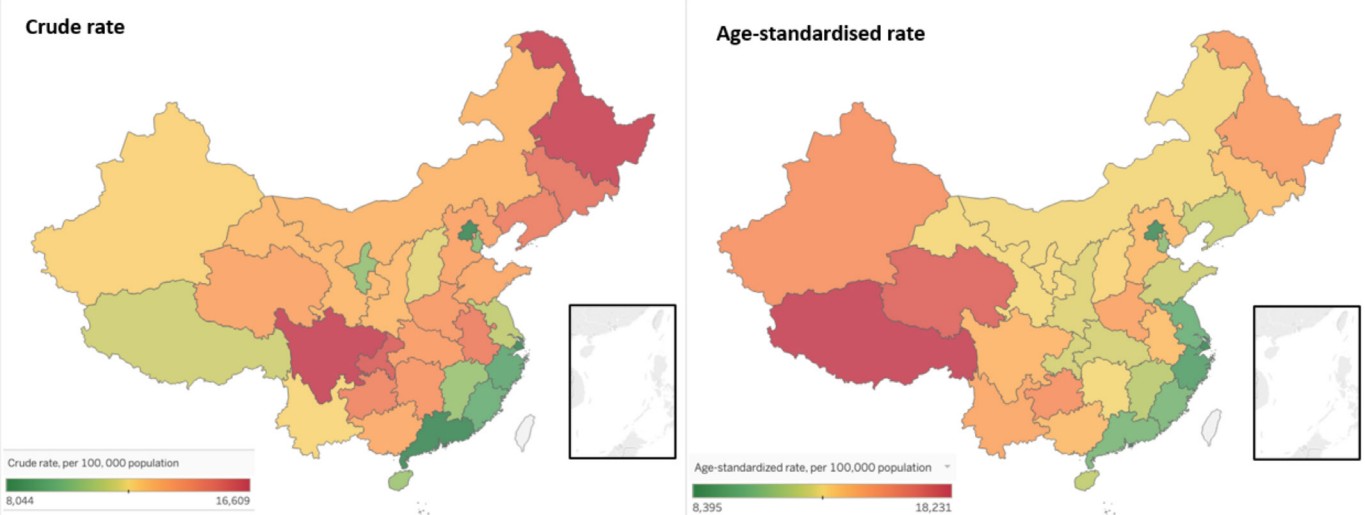

**Figure 1** The crude and age-standardised age-related diseases (ARDs) burden in 31 provinces of Mainland China, 2016. Data source: Global Burden of Diseases 2016. The ARDs burden was standardised using the Institute for Health Metrics and Evaluation standard population age structure (online supplemental appendix).

After standardising the age structure, we noted tiered gaps in the age-standardised ARD burden among the eastern, central and western provinces. Shanghai, Beijing, Zhejiang and Jiangsu had the lowest burdens, while the burdens of Tibet, Qinghai, Guizhou and Xinjiang were the highest. There were salient disparities between the sexes, with age-standardised ARD burden rates being 61.2% higher among males than females on average. This disparity was most apparent in neoplasms, where the burden rate was 1.5 times higher among males. We observed an inversion of this trend only among neurological disorders, where the burden rate was 10.9% higher among females.

In 2016, China's crude ARD burden-adjusted age was 50.38 years (95% UI: 49.91, 50.91), following the same regional disparity pattern as measured by DALYs (figure 2). Beijing, Guangdong, Shanghai, Zhejiang and Fujian, all located in eastern China, were the youngest, with a crude ARD burden-adjusted age below 40 years. In comparison, Sichuan, Heilongjiang, Chongqing, Jilin and Liaoning were the oldest, with a crude age close to or above 60 years. However, after the age structure across provinces was standardised, there were notable changes in the ARD burden-adjusted ages among several western provinces, especially for Tibet, Qinghai, Guizhou and Xinjiang. These four regions were the oldest by the standardised age, at 72.70 (95% UI: 60.53, 85.55), 66.82 (95% UI: 59.36, 74.03), 60.53 (95% UI: 53.76, 63.56) and 60.06 (95% UI: 53.00, 66.88) years, respectively. Shanghai, Beijing and Zhejiang still had the youngest age-standardised ARD burden-adjusted ages, all below 40 years.

Between 1990 and 2016, China's age-standardised ARD burden decreased despite the increasing life expectancy, with a national average decline of 34.6% compared with the 1990 burden. We found regional disparities in the magnitude of this decline. The largest reduction was seen in the eastern provinces, followed by the central and western provinces, with the average ARD burdens declining by 35.2%, 33.9% and 32.7%, respectively (figure 3). Breaking this down by province, Fujian experienced the most significant decline in the age-standardised ARD burden (42.2%), followed by Zhejiang (42.0%) and Beijing (40.7%). Guangxi, Qinghai and Guizhou experienced the smallest ARD burden decline in the same period, at 27.8%, 27.6% and 27.5%, respectively. The ARD burden declined more sharply in females (41.4%) than in males (29.8%). Between 1990 and 2016, the ARD burden had decreased for CVDs, neoplasms and CRDs, stayed almost unchanged for sense organ diseases and increased for neurological disorders (figure 4). CRDs experienced the sharpest decline (67.9%), followed by neoplasms (30.5%) and CVDs (26.8%). However, the burden of neurological disorders increased by 5% during the same period.

China's health investment grew between 2010 and 2016. The country's total health expenditures per capita increased from US$233.9 (exchange rate, US$1≈¥6.37) in 2010 to US$526.2 in 2016. The total health professional density (the number of total health professionals per 1000 people) increased by 38.6% from 4.4 in 2010 to 6.1 in 2016. In the same period, the licensed doctor density increased by 27.8%, while the licensed nurse density increased by 66.7% (online supplemental table S5).[39 44] We report the regression results in table 2. All coefficients of the health resource variables had the expected sign. The model results showed that neither higher health expenditures per capita nor higher health workforce density could significantly lead to a lower ARD burden, ceteris paribus. However, we found that the existing urban–rural gap in health workforce density was positively associated with the ARD burden, significant at the 10% level for all three indicators. A 100% increase in the urban–rural ratio in total health professional density,

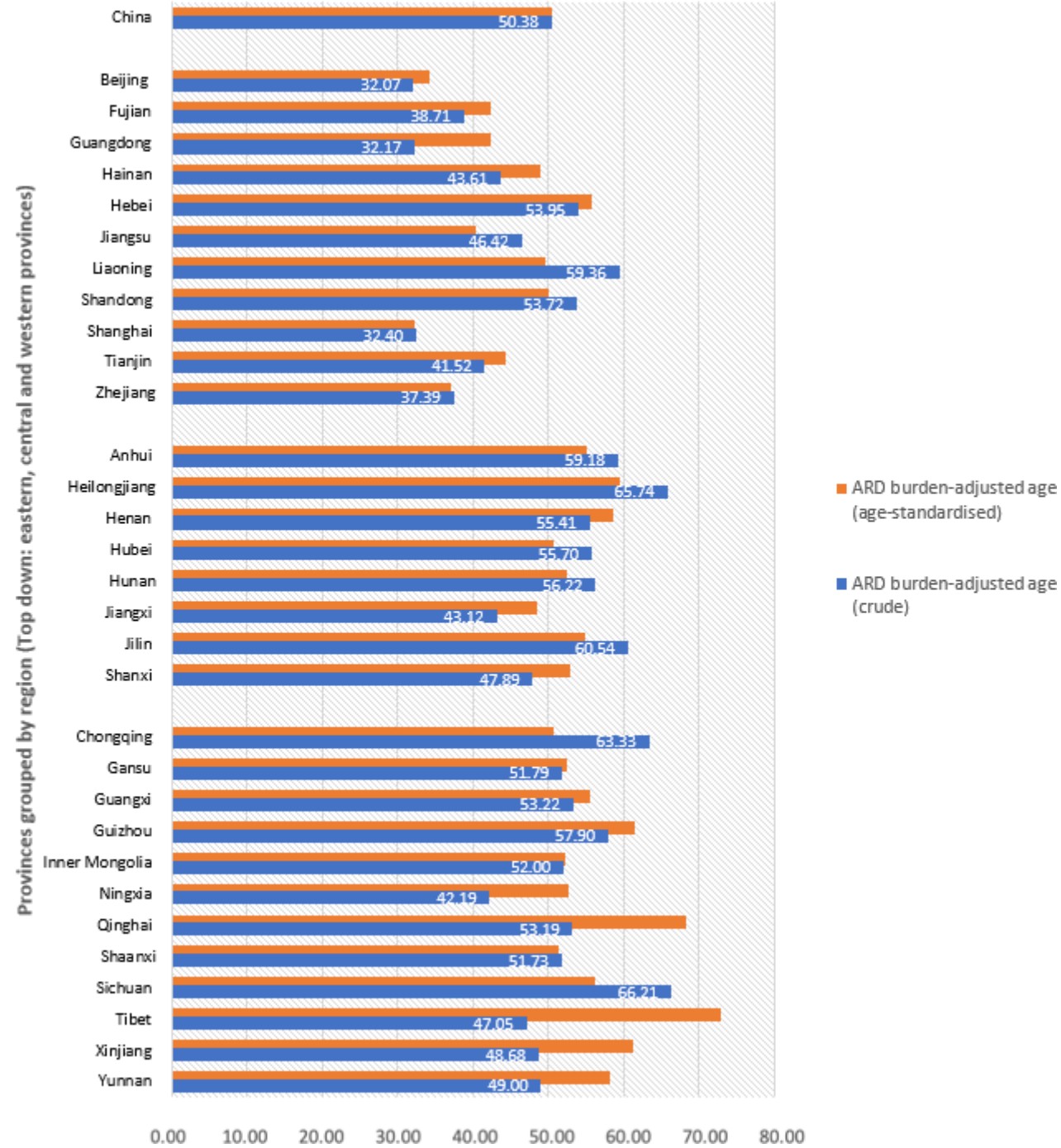

**Figure 2** The crude and age-standardised age-related disease (ARD) burden-adjusted ages of 31 provinces in Mainland China, 2016. Data source: Global Burden of Diseases 2016. The provinces were grouped into eastern, central and western provinces from top down. The age-related diseases burden was standardised using the Institute for Health Metrics and Evaluation standard population age structure (online supplemental appendix).

licensed doctor density and licensed nurse density led to 2.37% (95% CI: –0.41 to 5.15), 2.10% (95% CI: –0.27 to 4.46) and 2.02% (95% CI: –0.34 to 4.39) increases in the ARD burden, respectively, ceteris paribus.

## DISCUSSION
Healthy ageing has become a focal discussion topic in today's fast-ageing world, as good health in advanced age can provide continued opportunities for social and personal development.[2] The WHO officially proposed the concept of healthy ageing in 2015 and published the Global Strategy and Action Plan on Ageing and Health (2016–2020) in 2016.[2 45] The Decade of Healthy Ageing: Plan of Action (2020–2030) was issued in 2020 and asked for a whole-of-government and whole-of-society response to healthy ageing.[46] In response to this international call

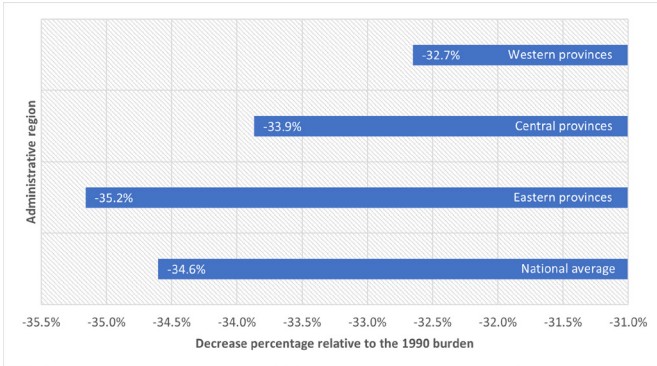

**Figure 3** Change of the age-standardised age-related diseases (ARDs) burden rates by administrative region in Mainland China, 1990–2016. Data source: Global Burden of Diseases 2016. The ARDs burden was standardised using the Institute for Health Metrics and Evaluation standard population age structure (online supplemental appendix).

and the needs of the world's largest ageing population, China has actively worked to address the shifts catalysed by an ageing society and promote healthy ageing.[47–49] We conducted this study to generate high-quality evidence to help China and similar countries be better equipped to face health challenges in ageing societies. To our knowledge, this is the first study that assesses the disease burden of ageing and its longitudinal changes in 31 provinces of Mainland China from 1990 to 2016. The analysis explored disparities in ARD burdens across regions, sexes and disease categories and used panel regression models to examine the impact of health resource indicators on the ARD burden.

Our findings underscore several key messages on the disease burden of ageing in China. First, NCDs account for over 90% of the country's total ARD burden. CVDs, neoplasms, CRDs, sense organ diseases and neurological disorders were the top five contributors to the ARD

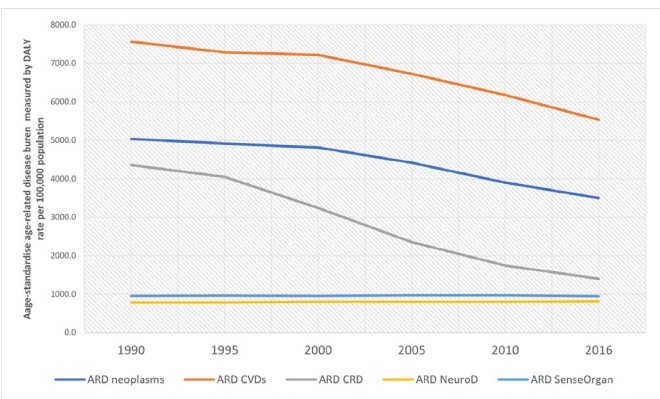

**Figure 4** Change of the age-standardised age-related diseases (ARDs) burden rates of the five leading ARD contributors in Mainland China, 1990–2016. Data source: Global Burden of Diseases 2016. The ARDs burden was standardised using the Institute for Health Metrics and Evaluation standard population age structure (online supplemental appendix). CRDs, chronic respiratory diseases; CVDs, cardiovascular diseases.

burden. Second, there are significant regional disparities in ARD burdens. The results show tiered differences in the ARD burden, with the lowest rates in the eastern provinces, followed by the central provinces, and the highest rates in the northeastern and western provinces. Notably, the crude ARD burden rate in Sichuan was 2.1 times that of Beijing in 2016, equal to a 34.14-year age gap measured by the ARD burden-adjusted age defined in this study. Third, several western provinces will face daunting ARD-induced challenges as their populations begin to age. After age structure standardisation, Tibet, Qinghai, Guizhou and Xinjiang had the four highest ARD burdens. However, in 2016, less than 10% of these provinces' populations were aged 65 years and older, compared with a national average of 10.8%.[30] Fourth, males are disproportionally affected by ARDs except for neurological disorders. Fifth, the overall age-standardised ARD burden has decreased since 1990, though the most significant decline was observed in the eastern provinces, followed by the central and western provinces. CRDs experienced the most significant decline in age-standardised burden by disease category, followed by neoplasms and CVDs. The burden rate of sense organ diseases has stayed almost unchanged since 1990. The burden of neurological disorders has increased since 1990, although by a small magnitude. Lastly, our results suggest that greater investment in health to reduce the urban–rural gap in the health workforce could help lower China's ARD burden.

Our study findings are consistent with previous studies that assessed the ARD burden globally and in China.[21 22] There is overall support for the idea that NCDs are the chief contributor to ARDs. As has been the case across the globe, the ARD burden has continuously decreased since 1990, declining most sharply in developed countries and regions. Disparities by sex are also similar; males are more negatively affected by the ARD burden and have experienced a lower historical decline. The magnitude of the burden measured in this study is slightly different from those in other studies due to differences across GBD datasets and the populations and diseases included. Our findings are novel in that we were able to measure the ARD burden and disparities across provinces in China and examine the impact of health resource indicators on the burden. Nevertheless, the regional disparities of age-standardised ARD burdens align with their health development status and are similar to the disparities in estimated HALE.[50–53] The rank of NCD burdens among the ARDs is consistent with that of the disease burden among older people in China.[34] Interestingly, we found that increasing health expenditures or workforce density does not significantly decrease the ARD burden, instead the key is to reduce the urban–rural gap in the health workforce density. Previous evidence has indicated that increasing health workforce density can improve health outcomes.[23 24] However, these studies focus on measuring health outcomes such as maternal mortality, infant and under-5 mortality as opposed to ageing-related outcomes. In addition, these studies did not control for geographic

**Table 2** Regression model results: assessing the impact of health resources on age-standardised age-related diseases burden in Mainland China, 2010–2016

| Independent variables | Model 1 (Total health professional density) | | Model 2 (Licensed doctor density) | | Model 3 (Licensed nurse density) | |
|---|---|---|---|---|---|---|
| | Coefficient (β) | 95% CI | Coefficient (β) | 95% CI | Coefficient (β) | 95% CI |
| Total health expenditures per capita | −2.29 | −6.46 to 1.88 | −2.02 | −6.18 to 2.14 | −2.32 | −6.56 to 1.91 |
| Health professional density | −1.34 | −3.48 to 0.79 | – | – | – | – |
| Health professional density: urban–rural ratio | 2.37* | −0.41 to 5.15 | – | – | – | – |
| Licensed doctor density | – | – | −0.93 | −3.30 to 1.44 | – | – |
| Licensed doctor density: urban–rural ratio | – | – | 2.10* | −0.27 to 4.46 | – | – |
| Licensed nurse density | – | – | – | – | −1.33 | −3.61 to 0.94 |
| Licensed nurse density: urban–rural ratio | – | – | – | – | 2.02* | −0.34 to 4.39 |
| GDP per capita | −0.16 | −5.09 to 4.77 | −1.00 | −6.34 to 4.33 | −0.34 | −5.57 to 4.89 |
| Population living in urban areas (%) | 1.82 | −15.60 to 19.24 | 1.83 | −15.13 to 18.80 | 3.03 | −14.95 to 21.01 |
| Female (%) | −0.60 | −2.50 to 1.30 | −0.87 | −2.66 to 0.92 | −0.83 | −2.80 to 1.14 |
| ≥Junior middle school education (%) | 3.25 | −1.66 to 8.16 | 2.30 | −2.25 to 6.84 | 3.03 | −1.72 to 7.79 |
| Adj. $R^2$ | 0.93 | – | 0.93 | – | 0.93 | – |

Data sources: Global Burden of Diseases 2016, Statistical Yearbook of China (2011–2017) and the Health Statistical Yearbook of China (2011–2017). All dependent and independent variables were transformed into natural logarithms for regressions except for the time dummies. The independent variables of interest and covariates were rescaled by dividing by 100. The coefficients can be interpreted as follows: every 100% increase in X can lead to a β% increase in Y. Model 1 used total health professional density (per 1000 population) as a proxy for human resources for health. Model 2 used licensed doctor density as a proxy (per 1000 population) for human resources for health. Model 3 used licensed nurse density as a proxy (per 1000 population) for human resources for health. Significance level: *p<0.10.
GDP, gross domestic product.

resource differences between rural and urban areas and were conducted in countries where resources were scarce. Thus, the context was quite different from that of the present study.

The Chinese State Council published 'The opinions on strengthening aged care work in the new era' in November 2021, outlining eight important action domains for addressing ageing. This document serves as a comprehensive, strategic plan to achieve healthy and active ageing in China.[49] The findings from the current study provide timely input into policy development for the governments of China to promote healthy ageing by reducing the disease burden of ageing. First, to reiterate the findings from previous studies, chronological age alone does not provide a strong enough basis for appropriate ageing resource planning or policymaking.[21 22] In China, provinces or municipalities with older populations, such as Shanghai, Beijing and Zhejiang, have much lower ARD burdens than some younger provinces, such as Tibet and Xinjiang. This phenomenon calls for careful consideration of underlying health burdens and potential threats to resource planning and policy design to address future ageing. Second, it is crucial to continue and strengthen

NCD prevention and control efforts, especially for CVDs, neoplasms, CRDs, sense organ diseases and neurological disorders. To prevent the continued increase in the ARD burden from neurological disorders, it is imperative to allocate additional health resources for the prevention, treatment and management of conditions such as Alzheimer's, dementia, Parkinson's and idiopathic epilepsy. Sense organ diseases cannot be ignored, especially those leading to blindness and vision loss, which can significantly damage the quality of life for the elderly. Third, the central government should continue to provide all-around support to the regions that face the greatest threat of high ARD burdens due to future ageing, especially the western provinces of Tibet, Qinghai, Guizhou and Xinjiang, to reduce regional disparities. Notably, that support should entail the establishment of a strong health workforce that can serve the local people. Fourth, aligned with the Rural Revitalization Strategic Plan issued by the State Council (2018–2022) in 2018,[51] the central and local governments should work collaboratively to strengthen the rural health system, particularly by increasing the health workforce density to reduce the urban–rural gap.

Our team have carefully compared the difference between estimates from the GBD 2016 and 2019 Studies to ensure the robustness of the findings from the current study. Methodologically, the GBD 2019 Study generally takes a similar approach as used in the GBD 2017 Study, which has enriched data input sources, modelling parameters, disease categories and upgraded calibration methods to get more accurate results compared with the GBD 2016 Study.[35 36] Our comparison results show that the crude ARD burden rate in 2016 based on estimates from GBD 2019 is slightly higher than estimates from GBD 2016, though the difference is within 8% and as low as 2% for neoplasms. Regarding the diseases included, we did not find significant changes at level three that would impact the results of the ARD burden. The findings are consistent with the comparison conducted by a previous study on the estimates generated by the GBD 2016 and 2017 Studies for the subnational level of China.[33] Therefore, subnational estimates of GBD 2016 can well represent the regional disparities of the ARD burden in China.

Our study has several limitations. First, the subnational analysis was based on the GBD 2016 Study estimates, which were not updated to 2019, as we do not have access to the GBD 2019 estimates at the subnational level of China. However, this was the best data source we could access, and as stated above, the results from the GBD 2016 Study are robust to show regional disparities after careful comparison. Second, our provincial analysis of the ARD burden focused on NCDs due to data availability, and we did not explore the burden of specific diseases. However, as NCDs accounted for over 90% of the ARD burden in China, these results can accurately represent existing regional disparities. Third, the study findings tend to underestimate the ARD burden, as GBD Study estimates fail to model the interactions between diseases, ignoring the burden caused by multimorbidity. Forth, though the definition used in the current study to measure the burden of ARDs is novel and easy to operationalise, it is only one of the many ways to measure it.[21 52–54] Future studies can gauge the pros and cons of different measurements and choose the most appropriate one for analysis. Fifth, we had to rely on interpolation to obtain annual estimates of burden rates from 2010 to 2016 for panel data analysis. However, the GBD research team used the same methods in their analysis.[38] In addition, the indicators for health resources are limited to total health expenditures and health workforce density, mainly due to data availability. Other important indicators, such as the efficiency of health funds utilisation and the quality of the health workforce, could be explored in future research when data become available.

## CONCLUSION

In conclusion, our study provides valuable and original insights into the disease burden of ageing in China and the existing regional disparities. Our results consistently show that an older demographic structure does not necessarily mean a heavier disease burden. Therefore, chronological age alone does not provide a strong enough basis for appropriate ageing resource planning or policymaking. Continuing to invest in population health by reducing the urban–rural gap in health workforce density would help reduce the disease burden of ageing in China. In so doing, the government of China needs to develop an effective policy and mechanism to attract more health professionals to work in rural and less developed areas. The governments of China, or other countries with similar demographic and disease profiles and development contexts, need to take urgent action to decrease the disease burden of ageing to create healthy ageing societies.

**Author affiliations**
[1]School of Risk and Actuarial Studies, UNSW, Sydney, New South Wales, Australia
[2]ARC Centre of Excellence in Population Ageing Research (CEPAR), UNSW, Sydney, New South Wales, Australia
[3]Social Policy Research Center, University of New South Wales, Sydney, New South Wales, Australia
[4]Department of Health Metrics Sciences, School of Medicine, University of Washington, Seattle, Washington, USA
[5]Institute for Health Metrics and Evaluation, Seattle, Washington, USA
[6]Global Health Research Center, Duke Kunshan University, Kunshan, China
[7]Duke Global Health Institute, Duke University, Durham, North Carolina, USA

**Acknowledgements** The authors thank the Institute for Health Metrics and Evaluation for making this study possible by sharing the Global Burden of Diseases 2016 Study estimates from China at the subnational level. The authors also appreciate the technical instructions from Dr Cheng Wan (UNSW) and Mr Lei Guo (Duke Global Health Institute) on drawing the maps.

**Contributors** SC designed the study under the supervision of KH, BL and HB, extracted the data, conducted the statistical analysis in Stata, developed the tables and figures and drafted the manuscript. SC and YS verified the data in the study. ST acted as gurantor and accepts full responsibility for the work and/or the conduct of the study, had access to the data, and controlled the decision to publish. All authors contributed to commenting, editing and approval of the final manuscript and accept the responsibility for submission to the journal for publication.

**Funding** This study received funding support from the Bill & Melinda Gates Foundation (OPP1148464), the UNSW, the ARC Centre of Excellence in Population Ageing Research and SHARP Fund UNSW (SHARP001).

**Map disclaimer** The inclusion of any map (including the depiction of any boundaries therein), or of any geographic or locational reference, does not imply the expression of any opinion whatsoever on the part of BMJ concerning the legal status of any country, territory, jurisdiction or area or of its authorities. Any such expression remains solely that of the relevant source and is not endorsed by BMJ. Maps are provided without any warranty of any kind, either express or implied.

**Competing interests** None declared.

**Patient and public involvement** Patients and/or the public were not involved in the design, or conduct, or reporting, or dissemination plans of this research.

**Patient consent for publication** Not applicable.

**Ethics approval** Not applicable.

**Provenance and peer review** Not commissioned; externally peer reviewed.

**Data availability statement** Data are available in a public, open access repository. Data may be obtained from a third party and are not publicly available. China's subnational-level data from Global Burden of Diseases 2016 were obtained from the Institute for Health Metrics and Evaluation under a private agreement. Researchers can apply for data access at https://www.healthdata.org/about/contact-us if interested in using the data. Other inputs data were obtained from open resources, including China Statistical Yearbook (http://www.stats.gov.cn/tjsj/ndsj/) and China Health Statistical Yearbook (http://www.nhc.gov.cn/wjw/tjnj/list.shtml). Researchers can access and download the data from these websites.

**ORCID iDs**
Shu Chen http://orcid.org/0000-0002-1108-3863
Yafei Si http://orcid.org/0000-0001-9334-8230
Katja Hanewald http://orcid.org/0000-0001-9932-2840
Bingqin Li http://orcid.org/0000-0003-2240-0876
Hazel Bateman http://orcid.org/0000-0002-6997-938X
Xiaochen Dai http://orcid.org/0000-0002-0289-7814
Chenkai Wu http://orcid.org/0000-0002-2256-0653
Shenglan Tang http://orcid.org/0000-0001-5727-078X

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
