## [Reviewer comments · BMJ Open]

ARTICLE DETAILS

TITLE (PROVISIONAL)	Disease Burden of Ageing, Sex and Regional Disparities, and Health Resources Allocation: A Longitudinal Analysis of 31 Provinces in Mainland China
AUTHORS	Chen, Shu; Si, Yafei; Hanewald, Katja; Li, Bingqin; Bateman, Hazel; Dai, Xiaochen; wu, chen kai; Tang, Shenglan

VERSION 1 – REVIEW

REVIEWER	Helen Fraser UCL, Institute of Healthy Aging
REVIEW RETURNED	04-Jul-2022

GENERAL COMMENTS	This article provides an interesting analysis of the burden of age-related diseases (ARDs) in China, summary of changes in ARD burden from 1990 to 2016, and exploration of how health resource allocation impacts the ARD burden. There are some recommended minor revisions, which I describe in the specific comments below. ----- Specific Comments ----- - Introduction/ Page 4 of 27: The terms "healthspan" (Zenin et al., 2019) and "lifespan" could be defined in the first paragraph, where the "years spent in good or bad health" is discussed.- Methods/ Page 5 of 27: There are many definitions of age-related diseases under different synonyms (Brody & Schneider, 1986; Brody & Grant, 2001; Kuan et al., 2021). Perhaps you could discuss why you selected the definition developed by Chang et al. (2019). Please add a reference for the sentence, "we defined the adult population as people aged 15 years and older". Please define "ARD burden-adjusted age".- Results/ Page 8 of 27: Although obvious, it is necessary to include the meaning of abbreviations. For example, CVDs (i.e., cardiovascular diseases), and CRDs (i.e., chronic respiratory diseases). The unit "years" should be included after ages, where appropriate.- Discussion/ p11 of 27: Another limitation is the broadness of disease categories and, therefore, specific diseases are not explored. For example, you have not included cataract, glaucoma, and age-related macular degeneration. These diseases are included in the GBD database.- Patient consent for publication/ p14 of 27: The sentence describing patient consent for publication needs revision.
--

REVIEWER	Zsombor Zrubka Óbuda University, HECON - Health Economic Research Center, University Research and Innovation Center
REVIEW RETURNED	21-Jul-2022

GENERAL COMMENTS	The authors present an analysis of age-related disease burden in China. Disparities by sex, province, time-related changes and determinants are explored. The article is well-written and uses reasonable methodology supported with the GATHER checklist. The research question is very complex, and therefore in my opinion the dependent variable and predictors require more elaboration and more careful exploration for spurious effects or methodological artefacts. The authors should also interpret carefully their findings, which were at times were not aligned with the supposed meaning of the key dependent variable. a) please elaborate and help the readers to interpret the ARD burden concept and results. The reader assumes that given DALYs are used for the calculation, with the ageing of the population, the mortality component decreases and disability component increases with absolute decrease of ARD burden. In the elaboration of results the authors interpret differently the results (p7 L46, p12, L13-21). b) please make reference to Chang et al. for the ARD-burden adjusted age calculation method. This concept is rather abstract, please add a few lines to explain what it means and how to interpret. c) the connection between predictor variables should be explored in greater depth. Total health expenditure may contain a large share of HCP costs, and the effect of rural/urban HCP density may depend on the proportion of rural/urban population. The correlation of these variables, potential for multicollinearity, and model diagnostics should be elaborated with greater detail for the regression models. d) I suggest that the authors also provide brief explanations about the choice of predictors, especially the urban/rural ratio should be explained in the context of the Chinese health system e) I suggest the authors reconsider their recommendation on p12 L17-18. The paper demonstrates that health expenditure does not affect ARD burden, while the authors suggest a continued investment. The recommendations should be more carefully aligned with results. I hope my comments help the authors to revise their manuscript.
--

VERSION 1 – AUTHOR RESPONSE

Reviewer: 1duid
Dr. Helen Fraser, UCL

Comments to the Author:

This article provides an interesting analysis of the burden of age-related diseases (ARDs) in China, summary of changes in ARD burden from 1990 to 2016, and exploration of how health resource allocation impacts the ARD burden. There are some recommended minor revisions, which I describe in the specific comments below.

Specific Comments

Introduction/ Page 4 of 27: The terms "healthspan" (Zenin et al., 2019) and "lifespan" could be defined in the first paragraph, where the "years spent in good or bad health" is discussed.

Thank you for this suggestion. We agree with the reviewer that providing a clear definition of the two terms can help the readers understand the necessity to measure the interactions between health and ageing. We have therefore revised the third sentence of the first paragraph, P3 L5-6, to "Although lifespans have increased substantially worldwide, it is unclear whether healthspans, i.e. the healthy and morbidity-free lifespan, have improved likewise" and added the Zenin et al. (2019) as one of the references.

Methods/ Page 5 of 27: There are many definitions of age-related diseases under different synonyms (Brody & Schneider, 1986; Brody & Grant, 2001; Kuan et al., 2021). Perhaps you could discuss why you selected the definition developed by Chang et al. (2019). Please add a reference for the sentence, "we defined the adult population as people aged 15 years and older". Please define "ARD burden-adjusted age".

Thank you for raising these very helpful and constructive comments. We are aware of the variety of definitions of age-related diseases you mentioned. We selected the definition developed by Chang et al. (2019) as it is practical for us to measure the age-related disease burden in the 31 mainland China provinces using the subnational estimates we can access. To address your comment, we have added in the *ARD selection and burden measurement* of the Methods section, P4 L44-46, that "Considering data availability, we followed the definition developed by Chang et al., to measure ARDs as diseases with incidence rates that increased quadratically with age among the adult population." In addition, we have added in the limitation of the discussion section, P12 L3-6, that "Forth, though the definition used in the current study to measure the burden of age-related diseases is novel and easy to operationalize, it is only one of the many ways to measure it. Future studies can gauge the pros and cons of different measurements and choose the most appropriate one for analysis."

We have revised the sentence "we defined the adult population as people aged 15 years and older" to "we focused on the population aged 15 years and older" (P5, L1-2). We focused on measuring disease burden among the population aged 15 years and older, as the estimates of the GBD study are aggregated by 5-year age groups. By doing so, we could include a more comprehensive list of diseases compared to focusing on the population aged 20 years and older.

To facilitate the understanding and interpretation of the "ARD burden-adjusted age", we further added the following definition in the 3rd paragraph of the *ARD selection and burden measurement* section, P5, L16, : "The ARD burden-adjusted age was defined as the equivalent age measured by the ARD burden."

Results/ Page 8 of 27: Although obvious, it is necessary to include the meaning of abbreviations. For example, CVDs (i.e., cardiovascular diseases), and CRDs (i.e., chronic respiratory diseases). The unit "years" should be included after ages, where appropriate.

We thank the reviewer for raising this comment. We have now added the abbreviations of CVDs and CRDs after their full name in the first paragraph of the results section when they first appeared in the manuscript (P6, L35). We have also added the unit "years" after ages where appropriate throughout the manuscript.

Discussion/ p11 of 27: Another limitation is the broadness of disease categories and, therefore, specific diseases are not explored. For example, you have not included cataract, glaucoma, and age-related macular degeneration. These diseases are included in the GBD database.

We agree with the reviewer that we did not explore the burden of specific diseases. However, diseases such as cataract, glaucoma, and age-related macular degeneration (GBD Level 4 causes)

were included under the category of blindness and vision loss (GBD Level 3 causes) of sense organ diseases (GBD Level 2 cause). We have therefore added in the limitations of the discussion section, P11, L47-48, that “Second, our provincial analysis of the ARD burden focused on NCDs due to data availability, and we did not explore the burden of specific diseases.”

Patient consent for publication/ p14 of 27: The sentence describing patient consent for publication needs revision.

We have revised it to “Not applicable” (same as other studies published on BMJ Open using GBD study estimates).

Reviewer: 2

Dr. Zsombor Zrubka, Óbuda University, Corvinus University of Budapest

Comments to the Author:

The authors present an analysis of age-related disease burden in China. Disparities by sex, province, time-related changes and determinants are explored.

The article is well-written and uses reasonable methodology supported with the GATHER checklist. The research question is very complex, and therefore in my opinion the dependent variable and predictors require more elaboration and more careful exploration for spurious effects or methodological artefacts. The authors should also interpret carefully their findings, which were at times were not aligned with the supposed meaning of the key dependent variable.

a) please elaborate and help the readers to interpret the ARD burden concept and results. The reader assumes that given DALYs are used for the calculation, with the ageing of the population, the mortality component decreases and disability component increases with absolute decrease of ARD burden. In the elaboration of results the authors interpret differently the results (p7 L46, p12, L13-21).

We thank the reviewer for this helpful comment. The current study estimates the burden of ARDs as another metric to measure population ageing, which takes into account both health and demographic ageing components.

The burden of ARDs, as measured by DALYs, has captured the years of life lost due to premature death (mortality component) and years of life lost due to time lived with disability (disability component). Specifically, one DALY can be interpreted as “the loss of the equivalent of one year of full health” (cited from WHO: <https://www.who.int/data/gho/indicator-metadata-registry/imr-details/158>). The historical change of the ARDs burden was assessed through comparing the change of age-standardized ARDs burden from 1990 to 2016, and our findings show the age-standardized burden of ARDs has decreased in China since 1990 (P7 L46, original submitted version). Nevertheless, we could not tell how the mortality and disability components of the burden have changed separately to co-drive the decrease.

In addition, we found out in the current study that the ARD burden is not necessarily higher in provinces with older demographic structure, such as Beijing and Shanghai. Such findings imply that “an older demographic structure does not necessarily mean a heavier disease burden and therefore chronological age alone does not provide a strong enough basis for appropriate ageing resource planning or policymaking” (P12, L13-21, original submitted version). This is one of the core messages that the current study conveys to readers, including policy makers as one important piece of evidence to help plan resources allocation for an ageing society.

b) please make reference to Chang et al. for the ARD-burden adjusted age calculation method. This concept is rather abstract, please add a few lines to explain what it means and how to interpret.

Thank you for pointing out this issue. The ARD-burden adjusted age is a novel concept developed by the current study. To help readers better understand this concept, we added in the third paragraph of the *ARD selection and burden measurement section* under Methods, P5, L16, that “The ARD burden-adjusted age was defined as the equivalent age measured by the ARD burden”, followed by calculation details (more to be found in the appendix), and its interpretation P5, L23, as “A younger ARD burden-adjusted age, therefore, implied a lower ARD burden.”

c) the connection between predictor variables should be explored in greater depth. Total health expenditure may contain a large share of HCP costs, and the effect of rural/urban HCP density may depend on the proportion of rural/urban population. The correlation of these variables, potential for multicollinearity, and model diagnostics should be elaborated with greater detail for the regression models.

We thank the reviewer for raising this issue and are fully aware that multicollinearity could undermine the robustness of the estimates of specific variables in a regression model. However, the results of the regression model as a whole are still valid and robust, as long as there is no perfect collinearity among regressors (one of the key assumptions about multivariate linear regression models).

Having said that, we performed regression models with fixed effects to check the relationship between 1) the total health expenditures per capita and health workforce density and 2) the rural/urban ratio in health workforce density and the proportion of rural/urban population. We have not detected statistically significant linear relationships among these variables before or after controlling for other covariates, including GDP per capita, sex and education (results in the appendix, Table S3 and Table S4).

Therefore, we added in the second paragraph of the *Analysing the impact of health resource allocation on the ARD burden* under Methods, P6, L11-12, that “We further explored the correlation of key variables to assess whether there was multicollinearity that could undermine the robustness of the estimates for specific variables in the regression model (appendix).” Detailed results of the multicollinearity check have also been added in the appendix Table S3 and Table S4.

d) I suggest that the authors also provide brief explanations about the choice of predictors, especially the urban/rural ratio should be explained in the context of the Chinese health system

Thank you for this very constructive comment. We fully agree that providing the context of the Chinese health system can help the readers better understand why we include urban/rural ratio into the regression models. Therefore, we have extended the second paragraph of the *Analysing the impact of health resource allocation on the ARD burden* section under Methods, P6, L2-7, as follows: “China has made remarkable achievements over the past decade in reducing the health disparities between urban and rural residents, especially through improving maternal and child health and extending health insurance coverage, among others, for its rural residents. However, there is still a noticeable urban-rural gap in health development, including access to quality health services, health workforce quantity and quality, and health outcomes.” Relevant references were also added in the manuscript.

e) I suggest the authors reconsider their recommendation on p12 L17-18. The paper demonstrates that health expenditure does not affect ARD burden, while the authors suggest a continued investment. The recommendations should be more carefully aligned with results.

We thank the reviewer for noticing this important policy implication emanating from the study findings. As the reviewer has kindly pointed out, we found that decreasing the urban-rural gap in the health workforce density was the key to reducing the ARD burden. Continuing investing in health is indeed essential, and the study findings highlight that the resources could be purposively allocated to help

strengthen the health workforce in rural areas, especially in the Chinese health system context where high-quality health human resources are largely concentrated in developed cities. Therefore, we now suggest on P12 L17-21 that “Continuing to invest in population health through reducing the urban-rural gap, especially in health workforce development, would be helpful to decrease the disease burden of ageing in China. In so doing, the Government of China needs to develop an effective policy and mechanism that can attract more health professionals to work in the rural and less developed areas”.

I hope my comments help the authors to revise their manuscript.

Once again, we thank the reviewer for providing these very helpful, constructive, and insightful comments, which we believe have substantially helped us improve our manuscript. We hope you will find the revised manuscript suitable for publication.

VERSION 2 – REVIEW

REVIEWER	Zsombor Zrubka Óbuda University, HECON - Health Economic Research Center, University Research and Innovation Center
REVIEW RETURNED	11-Sep-2022

GENERAL COMMENTS	Dear Authors, my comments were adequately addressed, congratulations to this interesting study. I have no further comments. I suggest a brief check for typos and minor English errors during proofing of the manuscript. I have no further comments.
--

VERSION 2 – AUTHOR RESPONSE

Reviewer: 2

Dr. Zsombor Zrubka, Óbuda University, Corvinus University of Budapest

Comments to the Author:

Dear Authors,

my comments were adequately addressed, congratulations to this interesting study.

I have no further comments. I suggest a brief check for typos and minor English errors during proofing of the manuscript.

I have no further comments.

Thanks for your time in reviewing this paper and helping improve its quality. We have carefully checked the paper again and corrected the remaining English errors and typos.